# Long Non-Coding RNA Signatures in Lymphopoiesis and Lymphoid Malignancies

**DOI:** 10.3390/ncrna9040044

**Published:** 2023-08-01

**Authors:** Hamed Baghdadi, Reza Heidari, Mahdi Zavvar, Nazanin Ahmadi, Mehdi Shakouri Khomartash, Mahmoud Vahidi, Mojgan Mohammadimehr, Davood Bashash, Mahdi Ghorbani

**Affiliations:** 1Department of Medical Laboratory Sciences, School of Allied Medical Sciences, AJA University of Medical Sciences, Tehran 1411718541, Iran; hamedbaghdadi2012@gmail.com (H.B.); mahmoud.vahidi@gmail.com (M.V.); mojganmehr20@yahoo.com (M.M.); 2Research Center for Cancer Screening and Epidemiology, AJA University of Medical Sciences, Tehran 1411718541, Iran; r-heidary@ajaums.ac.ir; 3Medical Biotechnology Research Center, AJA University of Medical Sciences, Tehran 1411718541, Iran; shakourmehdi@gmail.com; 4Department of Medical Laboratory Sciences, School of Allied Medical Sciences, Tehran University of Medical Sciences, Tehran 443614177, Iran; mahdi.zavvar@gmail.com; 5Department of Hematology and Blood Banking, School of Allied Medical Sciences, Shahid Beheshti University of Medical Sciences, Tehran 1985717443, Iran; ahmadi.n.1992@gmail.com

**Keywords:** long non-coding RNAs, lymphopoiesis, acute lymphoblastic leukemia, chronic lymphocytic leukemia

## Abstract

Lymphoid cells play a critical role in the immune system, which includes three subgroups of T, B, and NK cells. Recognition of the complexity of the human genetics transcriptome in lymphopoiesis has revolutionized our understanding of the regulatory potential of RNA in normal lymphopoiesis and lymphoid malignancies. Long non-coding RNAs (lncRNAs) are a class of RNA molecules greater than 200 nucleotides in length. LncRNAs have recently attracted much attention due to their critical roles in various biological processes, including gene regulation, chromatin organization, and cell cycle control. LncRNAs can also be used for cell differentiation and cell fate, as their expression patterns are often specific to particular cell types or developmental stages. Additionally, lncRNAs have been implicated in lymphoid differentiation, such as regulating T-cell and B-cell development, and their expression has been linked to immune-associated diseases such as leukemia and lymphoma. In addition, lncRNAs have been investigated as potential biomarkers for diagnosis, prognosis, and therapeutic response to disease management. In this review, we provide an overview of the current knowledge about the regulatory role of lncRNAs in physiopathology processes during normal lymphopoiesis and lymphoid leukemia.

## 1. Introduction: A Glance into Long Non-Coding RNAs

Lymphocytes are a heterogeneous population of blood cells consisting of three main types: B, T (including Th1, Th2, Th17, T regulatory, and T cytotoxic), and NK cells [1]. Lymphopoiesis is a term used to describe the process of the creation of lymphoid cells, which plays a crucial role in the immune system. Lymphocyte differentiation pathways that give rise to diverse mature lymphocytes are regulated by a set of genes that are expressed and repressed in complexes [2]. Genomic alterations are directly involved in the abnormal proliferation of mature and immature lymphocytes, which causes chronic and acute leukemia [3].

Non-coding RNAs (ncRNAs) are functional single-stranded RNA molecules that are not translated into proteins, although there are exceptions where ncRNAs are translated into peptides. NcRNAs have recently been identified as playing a critical role in a wide range of biological processes. Despite this, a large proportion of ncRNAs remain to be characterized, and their functional roles remain unknown. This suggests that the majority of the human genome serves a purpose beyond encoding proteins. This makes it critical to understand the role of these ncRNAs to fully grasp the complexity of the human genome. This implies that they are involved in a wide range of cellular activities, such as cell differentiation, development, and response to environmental changes. Understanding their role in these processes is essential to further our knowledge of the human genome [4,5,6].

Non-coding RNAs can be divided into two categories: linear ncRNAs and circular ncRNAs. According to their length, linear ncRNAs are also divided into two subgroups including long non-coding RNAs (lncRNAs) and small ncRNAs (sncRNAs). Each of these categories plays a crucial role in a variety of biological processes, such as epigenetic regulation, chromatin remodeling, gene expression regulation, controlling cell differentiation, and modulating the immune system [4,7]. LncRNAs are greater than 200 nucleotides long and have a wide range of functions. LncRNAs are an important class of non-coding RNAs that have been discovered in recent years. They have been implicated in a wide range of cellular processes and in the development of various diseases, including cancer, immune diseases, diabetes, and cardiovascular diseases [8,9,10,11,12]. According to the Human GENCODE statistics, the lncRNAs of the human genome are estimated to be approximately 19,000 genes [13], most of which are transcribed by RNA polymerase II and often have a polyadenylated tail at their 3′ ends and 7-methyl guanosine (m7G) caps at their 5′ ends [14]. There are three main categories of lncRNAs based on their functions: (1) functional lncRNAs, (2) non-functional lncRNAs caused by transcription noise, and (3) lncRNAs whose transcripts are not necessary and transcription alone suffices to exert their effects [4]. Figure 1 illustrates how lncRNAs are synthesized from DNA and processed before being exported to the cytoplasm. In the final step, the lncRNA is transformed into a mature form that can be used by the cell.

Several biological processes depend on lncRNAs’ fine-tuned functions, including epigenetic regulation, chromatin remodeling, DNA replication, mRNA splicing, and translation. Their overexpression, mutation, or deficiency can affect cellular statuses and cause a variety of diseases [4]. According to the mechanisms of gene expression regulation, lncRNAs can be categorized as guide, signaling, decoy, enhancer, and scaffolding RNAs. Guide lncRNAs regulate gene expression by targeting and recruiting transcriptional complexes to specific genes. Signaling lncRNAs are involved in cell signaling pathways and can act as transcription factors. Decoy lncRNAs act as sponge mRNA and transcription factors, inhibiting them from binding to the intended target. Enhancer lncRNAs activate the transcription of specific target genes by binding to enhancer elements and recruiting transcription factors. Scaffolding lncRNAs can bring together proteins involved in transcriptional regulation and help maintain their interactions [15,16]. LncRNAs are involved in both the oncogenesis and the tumor suppressor pathways. Oncogenic lncRNA molecules are crucial in cancer development and progression. This includes the activation of oncogenic pathways, the activation of cancer-promoting genes, and the stabilization of oncogenic proteins. These lncRNAs also play a significant role in cell growth and proliferation, as well as cancer metastasis. Tumor suppressor lncRNAs, on the other hand, suppress tumor growth by either inhibiting the expression of oncogenes or inducing the expression of tumor suppressor genes. In addition, these lncRNAs are involved in cell cycle progression, apoptosis, and angiogenesis [7].

In recent years, lncRNA has been found to play a key role in determining a cell’s fate and orchestrating its response to stimuli. Thus, lncRNAs have become a major focus in the study of the regulation of immune cells and lymphoid cells, including B, T, and NK cells. LncRNAs have been shown to be important for the development and function of lymphoid cells. It can be involved in the regulation of various immune pathways, including those involved in inflammation, differentiation, proliferation, and apoptosis of lymphoid cells. In addition, it regulates cytokine secretion, cytokine signaling, and antigen presentation. Considering the expression pattern of lncRNAs during lymphoid differentiation and their roles in lymphoid leukemias, a growing list of relevant lncRNAs has been identified in lymphoid leukemogenesis in recent years. Interestingly, a distinctive lncRNA expression signature has been associated with leukemia prognosis. Moreover, this lncRNA expression signature has been linked to treatment response and relapse, providing further insight into the disease’s progression. As a result of these findings, lncRNAs can be used to modulate the development of immune-related diseases. It appears that these genes may be used for either prognostic or therapeutic purposes [17].

Here, we summarize the current knowledge about lncRNA expression during normal lymphopoiesis and the consequences of altered lncRNA expression in lymphoid leukemias. However, few studies have investigated the role of these molecules in lymphopoiesis.

## 2. LncRNAs in Normal Lymphopoiesis

LncRNAs are involved in the regulation of hematopoietic lineage commitment, self-renewal, proliferation, and differentiation of hematopoietic stem cells (HSC), as well as controlling their fate and quiescence. In addition to acting as enhancers and suppressors of transcription, these lncRNAs have the potential to affect the expression of a wide variety of genes involved in HSC differentiation, hematopoiesis, and lymphopoiesis [18,19]. As a consequence, lncRNAs may be important regulators of cell differentiation and maturation in the hematopoietic system.

At the HSC stage, many lncRNAs are essential for the proper regulation of HSC differentiation to multipotent progenitors (MPPs) and are key components of the regulatory system. These include H19, HOTAIR, XIST, lncHSC-1, and lncHSC-2 lncRNAs, which are expressed in long-term HSCs and control HSCs quiescence and self-renewal. H19 expression is downregulated during HSC differentiation to MPPs. It is known to act as a negative regulator of the Wnt/β-catenin pathway and is essential for the activation of the hematopoietic transcription factor PU.1, which is important for HSC differentiation [20,21].

MEG3 lncRNA is highly expressed in HSCs and suppressed in the MPP stage. It plays a key role in preserving the functionality of HSCs by suppressing PI3K/AKT/mTOR signaling [22]. HOTTIP and MALAT1 lncRNAs have been found to be upregulated and downregulated during HSC differentiation to MPP, respectively. These lncRNAs are known to be involved in the regulation of transcription factors such as E2F1, GATA2, and c-Myc [23,24].

LncRNAs have recently been found to play an important role in HSC differentiation into common lymphoid progenitor (CLP) cells. Studies have shown that lncRNAs are able to promote the development of CLPs by modulating the expression of essential transcription factors, such as GATA2 and E2A, which are important for CLP development [25,26].

During the progenitor stage, lncRNA expression patterns can distinguish cells of the B and T lineages [25]. This means that progenitor cells committed to B or T cells express different lncRNA levels. This could be useful for understanding how the B and T lineages differentiate and how to engineer cells to specialize in a certain lineage. Understanding the differences in lncRNA expression patterns between the B, T, and NK lineages can also provide insight into how to differentiate cells based on their lineage.

Figure 2 provides a better overview of the role of lncRNAs at different stages of normal lymphopoiesis.

### 2.1. LncRNAs in Normal B Lymphopoiesis

There have been several studies investigating lncRNA expression patterns at different stages of B-cell development [25,26,27,28,29]. A significant difference in lncRNA expression has been found between B-cell subsets as they mature and develop [28]. The transcriptomes of 11 human B cell subsets revealed that several lncRNAs are expressed during B cell development [26]. In early B-cell development, lncRNAs including LEF1-AS1, SMAD1-AS1, MYB-AS1, and CTC-436K13.6 are associated with genes such as RAG2, VPREB1, DNTT, LEF1, SMAD1, and MYB. Brazao et al. reported that 784 lncRNAs of pro-B cells and 717 lncRNAs of mature B cells overlapped or were located less than 1 kb downstream of the PAX5 binding sites (a transcription factor necessary for B-cell development); evidence shows that 109 of these lncRNAs are regulated by PAX5, indicating their importance in B-cell homeostasis [30]. In agreement, a study in mice proposed a dominant role of germ-line transcribed lncRNA PAX5-activated intergenic repeat (PAIR) elements, PAIR4 and PAIR6, during V(D)J recombination in progenitor B cells [31]. It has also been suggested that transcription factor YY1, which is involved in rearranging the distal VH genes and transitioning precursor B cells, interacts with and relocates lncRNA Xist to inactivated X-chromosomes in activated B cells, thereby altering gene regulation on X-linked chromosomes compared to naive B cells [32]. During the proliferative stages of B-cell development (preBI, preBII, centroblasts, and centrocytes), mitotic cell cycle-related genes such as KIF23, PLK4, and CENPE are associated with the lncRNAs OIP5-AS, MME-AS1, and CRNDE [33,34,35].

The lncRNA XIST has been found to be involved in the regulation of B-cell proliferation. XIST is expressed in mature B cells, and its upregulation is associated with the proliferation of B cells. Additionally, XIST can regulate the expression of genes involved in B-cell proliferation, such as CDKN2A and BTG2 [32]. LINC00926 lncRNA is linked with the CD22 B cell receptor, which is known for B cell signaling and survival [36].

The lncRNA DLEU2 has been found to be involved in the regulation of B-cell proliferation by regulating the expression of a gene called MYC. In particular, it has been demonstrated that DLEU2 is able to bind to the promoter region of MYC and downregulate its expression, which in turn leads to decreased B cell proliferation [37].

PVT1, LINC00487, LINC00877, RP11-203B7.2, and RP11-132N15.3 lncRNAs are associated with the AID and SERPINA9: two genes specifically found in centroblasts and centrocytes [26].

A transcriptome analysis of B-cell subtypes in mice revealed 4516 lncRNAs that were differentially expressed during B-cell maturation and activation. A strong positive correlation was found between LNCGme022323 and the lightpink4 gene cluster, which contains 31 protein-coding genes regulated in marginal zone B cells [30]. Tschumper et al. reported that MIAT, MALAT1, and LINC00152 lncRNAs increased expression during plasma cell development [38].

Altogether, despite some limitations like the restricted number of B-cell subsets included and the use of microarrays that prevent novel transcript identification in recent studies, these studies provide valuable insights into B cell subset-specific lncRNA transcripts. The field is now open for studies aiming to establish their roles in normal B-cell development [39].

### 2.2. LncRNAs in Normal T Lymphopoiesis

Microarray analysis has confirmed that lncRNAs might be critical molecular determinants in CD4+ T-cell activation and differentiation [40]. Currently, IFNγ-AS1 lncRNA (also termed NeST or TMEVPG1) is known as a lineage-specific lncRNA and is transcribed on the opposite DNA strand to the IFNγ, contributing to its expression [41]. Naive CD4+ T cells do not express IFNγ-AS1 lncRNA, but this lncRNA is increased by STAT4 and T-bet transcription factors in response to Th1 differentiation signals. Inflammatory cytokines have been shown to be significantly increased when IFNG-AS1 is specifically overexpressed in T cells [41,42]. There is also a lncRNA known as linc-MAF-4 that is involved in Th1/Th2 polarization [28,43]. It is a chromatin-associated lncRNA that is specific to the Th1 subset and promotes Th1 differentiation by suppressing the expression of the Th2 transcription factor MAF. As a result, the downregulation of linc-MAF-4 causes an increase in the expression of GATA3 and IL4, which induce T-cell differentiation toward the Th2 cell lineage [28]. The lncRNA GATA3-AS1 promotes Th2 cell development through the SATA3 pathway, and it is necessary for the expression of IL5 and IL13 and the expression of GATA3 during Th2 cell commitment [44].

Several long noncoding RNA genes play a role in the early differentiation of T helper subsets, and these lncRNAs are correlated with their neighboring protein-coding genes [45]. In this regard, TH2-LCR lncRNA is co-expressed with neighboring gene clusters that encode Th2 cell cytokines (IL4, IL5, and IL13), regulating their expression [46]. Similarly, Koh et al. reported that depletion of the TH2-LCR lncRNA cluster inhibits WDR5 recruitment to IL4, IL5, and IL13 genomic promoters and blocks H3K4 trimethylation in these regions, causing a decrease in the expression of IL4, IL5, and IL13 [47]. Xia et al. reported a comprehensive analysis of the lncRNA status during CD4+ T-cell development and activation (Figure 2) [40].

Another lncRNA that affects T-cell differentiation is lnc-ITSN1-2. By targeting miR-125a and upregulating IL-23R, this lncRNA increases proliferation and activation of CD4+ T cells and promotes differentiation to Th1/Th17 cells [48].

As a Th2-specific lncRNA, lincR-Ccr2-5′AS, together with the GATA-3 transcription factor, regulates the expression of immune genes and plays a critical role in the chemokine signaling pathway of the Th2 subset. As reported by Hu et al., this lncRNA regulates transcription of Th2 key chemokine gene clusters (CCR1, CCR2, CCR3, and CCR5), and its depletion impairs migration of these cells to the lung after in vivo transfer [49].

Additionally, lncRNAs are involved in the differentiation of Th17 cells through the interactions with the transcription factor DDX5 (DEAD-box RNA helicase), which acts as a partner of the transcription factor RORγt in Th17 cells [50,51]. Rmrp lncRNA promotes the RORγt–DDX5 complex assembly at the genomic loci of genes encoding main Th17 cell effector molecules, such as IL17a and IL17f [52]. In addition, Zhang et al. demonstrated that knockdown of lncDDIT4 in naive CD4+ T cells promotes the differentiation of Th17 cells by increasing DDIT4/mTOR pathway activation [53].

Different lncRNAs are involved in Treg activation and differentiation. The lncRNA FLICR (Foxp3 long intergenic noncoding RNA) is expressed only in mature Tregs and inhibits differentiation by decreasing Foxp3 expression [54]. Zemmour et al. reported that the roles of IL-2 and FLICR in Treg homeostasis are diametrically opposite; when IL2 levels are low, Flicr reduces the chromatin accessibility of the Foxp3 locus. On the other hand, the expression of Foxp3 is stabilized in the absence of FLICR [55]. A novel lncRNA, Flatr (Foxp3-specific lncRNA anticipatory of Tregs), was identified as a potential regulator of Tregs. According to this study, downregulation of Flatr lncRNA can decrease Treg induction by contributing to differentiated T cells [56].

The role and function of lncRNAs in CD8+ T-cell development and function is still poorly understood. Kanbar et al. reported that the lncRNA Malat1 is a regulator of CD8+ T-cell differentiation, and MALAT1 knockdown significantly reduced T effector cell differentiation at the peak of infection [57]. Upregulation of programmed cell death protein 1 (PD-1) prevents excessive T-cell activation. It was reported that different levels of PD-1 expression were observed in activated CD8+ T cells, with different expression of LncNDEPD1. miR-3619-5p destabilizes PD-1 mRNA, and lncNDEPD1 sponges this miR, causing PD-1 expression to be upregulated [58].

### 2.3. LncRNAs in Normal NK Lymphopoiesis

An analysis of the lncRNA profiles of different NK cell populations identified several novel lncRNAs, including lnc-CD56, a NK cell-specific lncRNA that positively regulates CD56 expression. Zhang et al. reported that lnc-CD56 has been shown to positively regulate CD56 in primary NK cells and has a key role in the differentiation of CD34+ progenitor cells to NK cells. They also reported that the knockdown of lnc-CD56 led to a decrease in CD56 expression. These data indicate that lnc-CD56 plays an essential role in NK cell differentiation [59]. This lncRNA acts as a cis regulator in the NK cell. Initially, there was a hypothesis that lnc-CD56 targets shRNA directly bound to CD56 pre-mRNA, leading to its degradation and reducing mRNA levels; however, this hypothesis was rejected, and its mechanism has not yet been determined. Interestingly, catRAPID predicted that lnc-CD56 has interactions with critical transcription factors of NK cells like IRF2, TBX21, IKZF2, ELF4, and EMOES [59]. Thus, additional studies of lnc-CD56 in human NK cells will be required to confirm its roles in NK cell biology. Additionally, evidence shows that IFNγ-AS1 is expressed not only in CD4+ Th1 and CD8+ T cells but also in NK cells [41,49,60]. IFNγ-AS1 expression in NK cells is related to its upregulation upon cellular activation, contributing to increased IFNγ production. IFNγ-AS1 can join the web of intracellular signaling by following signals received through their activating and inhibitory receptors, such as the signaling molecules DAP12, ZAP70, and Syk [61].

## 3. Alterations in lncRNAs in LYMPHOID Malignancy

In recent years, increasing knowledge about lncRNAs that are abnormally expressed or mutated in leukemia has further highlighted the importance of these molecules in leukemogenesis. Some lncRNAs are deregulated in all leukemias, such as NEAT1, MALAT1, and GAS5 [62]. Furthermore, a distinctive lncRNA expression signature associated with leukemia prognosis has suggested the possible usage of these genes either in prognostic or treatment decisions [17]. In the following sections, we present an overview of current knowledge regarding the expression and function of lncRNAs in lymphoid malignancies.

### 3.1. Alterations in lncRNAs in Acute Lymphocytic Leukemia (ALL)

It has been reported that lncRNAs can play a role in the pathogenesis of ALL by affecting the immune system. For instance, the lncRNAs NONHSAT027612.2 and NONHSAT134556.2 alter two key components of the innate immune system, including TLR4 and NOD2. Of note, these two molecules showed higher levels of expression in children with ALL than in controls [63]. Through a bioinformatics analysis, it was found that uc.112 has a higher expression in T-ALL than in B-ALL. Interestingly, the results showed that the genome of this lncRNA has binding sites for several transcription factors such as PAX5, IRF4, EBF1, and RUNX1, further highlighting the potential role of uc.112 lncRNA in disease pathogenesis [64]. Moreover, an investigation of circPVT1 function in adult ALL patients revealed that this lncRNA can inhibit proliferation and induce apoptosis by regulating neighboring C-MYC and BCL2 genes [65].

Apart from pathogenesis, lncRNAs also show different expressions at different stages of the disease. In this regard, RP-117C120C, whose high levels are associated with high expression of cCD79a, increases in newly diagnosed patients, while decreasing in the complete remission phase [66]. Their different expressions may also affect their response to treatment. Gasic et al. showed that GAS5 reacts with glucocorticoid receptors and therefore could be a potential marker of pharmacotranscription [67]. The effect of lncRNAs on the glucocorticoid response in ALL has been also reported in other studies. For example, HOXA-AS2, whose overexpression is induced by the transcription factor TCF7L2, increases resistance to glucocorticoids [68]. MALAT1 and NEAT1 are other lncRNAs that seem to have an effect on treatment, as they showed higher expressions in the group of MRD-positive patients compared to MRD-negative patients [69].

#### 3.1.1. Alterations in lncRNAs in B-ALL

By conducting bioinformatics studies on newly diagnosed and relapsed pediatric and adult patients, James and colleagues identified 1235 dysregulated lncRNAs associated with each subgroup of B-ALL. They also found 942 lncRNAs related to relapse and methylation profiles. They identified a strong correlation between Ph-like-specific lncRNAs and the genes involved in the activation of the PI3K-AKT, mTOR, and JAK-STAT signaling pathways. Moreover, there was a significant correlation between the DUX4-specific lncRNAs and the genes associated with the activation of the TGF-β and Hippo signaling pathways. Furthermore, relapse-specific lncRNAs were correlated with the activation of metabolic and signaling pathways [70]. In another study, Cuadros et al. compared the lncRNAs profile of children with or without t(12;21) translocation. While TCL6 and TCL1B were the most upregulated lncRNAs, relapsed or expired patients had low levels of TCL6 [71]. Accordingly, a study on pediatric patients with the same translocation showed that lnc-NKX2-3-1, lnc-TIMM21-5, ASTN1-1, and lnc-RTN4R-1 were regulated by this fusion protein [72]. Similarly, Lajoie et al. showed high expression of BALR-1 and LINC0098 lncRNAs in pre B-ALLs with t(12;21). Interestingly, these two lncRNAs were associated with hyperdiploidy [73]. Similar to these studies, Fernando et al. demonstrated the association between lncRNA expression patterns and cytogenetic abnormalities in three B-ALL subgroups, including t(12;21), t(1;19), and MLL-rearrangement. Their results indicated that an increased level of BALR-2 is associated with prednisolone resistance and poor survival [74].

While some lncRNAs act as tumor suppressors, others play oncogenic roles in B-ALL. Evidence of a tumor-inhibitory role played by lncRNAs was provided by the study by Wang et al., who examined Philadelphia-positive leukemia patients. The authors showed that IUR lncRNA in humans and mice increased after treatment with imatinib, and knocking down this molecule could increase tumor survival and growth. It has been suggested that this lncRNA inserts its tumor suppressor activity by inhibiting STAT5 and CD71 [75]. LINC00221 has also been shown to have anti-proliferative and pro-apoptotic effects by binding to miR152-3P and upregulating ATP2A2 [76]. A mechanism that cells use to preclude tumorigenesis is the protective response to DNA damage. In this regard, Gioia et al. identified three lncRNAs, RP11-624C23.1, RP11-203E8, and RP11-446E9, which provide an advantage for tumors when silenced [77]. Fernando et al. also showed that CASC15 could act as a tumor suppressor by regulating the SOX4 transcriptional factor in leukemia patients with RUNX1 rearrangement. Notably, elevation of this lncRNA increases the death of malignant cells following prednisolone treatment [78].

Surprisingly, a study of the AML population considered a proto-oncogenic role for CASC15, as a mutation in IDH could induce this lncRNA [79]. In agreement with the oncogenic roles of lncRNAs, a study found that RP11-137H2.4 knockdown inhibited proliferation and restored glucocorticoid sensitivity in children with B-ALL [80]. Similarly, López et al. showed that high expression of LINC00152 has a significant relationship with the recurrence of disease in B-ALL patients. This molecule regulates genes involved in cell-substrate adhesion and peptidyl-tyrosine autophosphorylation [81]. A study in patients with MLL-rearrangement led to the identification of a group of dysregulated lncRNAs, including ENST00000443469. Notably, ENST00000443469 and NR_033375 were found to induce proliferation while inhibiting apoptosis in the MLL-rearrangement positive cell lines [82]. In some cases, certain variants of lncRNAs can be involved in the pathogenesis of the disease. A study of 213 children with B-ALL led to the introduction of a specific allele in an lncRNA known as ANRIL (also known as CDKN2B-AS1), which increases susceptibility to leukemia [83]. Similarly, a study by Rosa et al. showed that variant 3 of the ZNF695 transcript in children with B-ALL is beneficial to the tumor and predicts patient survival significantly [84]. An overview of the most important lncRNAs associated with B-ALL are summarized in Table 1.

#### 3.1.2. Alterations in lncRNAs in T-ALL

Through comprehensive RNA-Seq profiling, an accumulated number of studies have indicated that lncRNAs play critical roles in T-ALL pathogenesis [89,90,91,92]. In this regard, Yang et al. reported that SNHG1 is upregulated in T-ALL under in vivo and in vitro conditions and has a tumor suppressor effect through interactions with miR 124-3P. Moreover, it has been reported that ARIEL lncRNA is upregulated in TAL1+ T-ALL patients, increasing tumor progression through ARID5B activation [93]. Several studies have also indicated the association between Myc and lncRNAs in lymphoid neoplasia [94,95,96]. In this regard, lncRNA PVT1, which has a stabilizing role for Myc protein, induces its tumorigenic activity in ALL [97]. TLX1 is another transcription factor that controls several important lncRNAs in T-ALL subtypes [98]. Among T-ALL-related lncRNAs, NOTCH-driven long noncoding RNAs have attracted great attention in recent years. Durinck [89] and Trimarchi [99] reported that LUNAR1, which is highly regulated by NOTCH, leads to T-ALL progression by increasing the levels of IGF1R. The upregulation of both NALT lncRNA [100] and NEAT1 lncRNA [101] can serve as oncogenic molecules in T-ALL.

Some lncRNAs play their role by employing other molecules. According to the study conducted by Singh et al. on the T-ALL cell line, the H19 lncRNA induces SOX2, OCT4, and NANOG transcription factors through PIM kinase [102]. Likewise, Renou et al. implied that TRAF6 is regulated through sponging miR-335-3P by the lncRNA CDKN2B-AS1 [103]. Li et al. revealed that the lncRNA AWPPH interacts with the ROCK2 oncogene in pediatric T-ALL to inhibit apoptosis [104]. Another anti-apoptotic lncRNA is T-ALL-R-LncR, whose knockdown facilitates the formation of the Par-4/THAP1 complex, which in turn upregulates proapoptotic Smac by activating caspase-3 [105].

The miR-655-3p has an inhibitory role in proliferation, invasion, and migration of some cancer cells. Yang et al. reported upregulation of lncRNA EBLN3P expression in clinical samples of T-ALL patients, which sponges miR-655-3p and prevents the inhibitory effect of miR-655-3p on the proliferation, invasion, and migration of leukemic cells [106]. An overview of the most important lncRNAs associated with T-ALL is detailed in Table 2.

### 3.2. Alterations in lncRNAs in Chronic Lymphocytic Leukemia (CLL)

As in ALL, the expression of some lncRNAs is associated with distinct cytogenetic abnormalities, and similar associations have been reported in CLL. According to a recent study, the highest levels of Lnc-IRF2-3 and Lnc-ZNF667-AS1 were found in CLL patients with del17p and del11q, respectively. Of note, increased expression of these lncRNAs was associated with poor prognostic factors, including high levels of ZAP70 protein and un-mutant IGHV [113]. In contrast, lncRNA is associated with ZAP70 protein levels and un-mutant IGHV. Indeed, this lncRNA protects the DNA of malignant cells from chemotherapy-induced damage, leading to a poor prognosis [114]. Furthermore, it has been reported that high levels of MIAT are associated with advanced CLL based on chromosomal abnormalities [115].

Apart from MIAT, which is an oncogenic lncRNA that protects leukemia cells against apoptosis, several lncRNAs with tumor-inhibitory roles have been reported in CLL as well. For example, a study found that GAS5 plays an inhibitory role in tumorigenesis and metastasis by sponging miR-222 [116]. It has also been indicated that GAS5 can suppress MYC expression, suggesting another mechanism through which GAS5, at least partly, induces its tumor-suppressive effect [96]. An examination of 80 newly diagnosed patients with CLL showed that low levels of LincRNA-P21 were significantly associated with poor outcomes, and that this lncRNA showed a negative correlation with CD38, ZAP70, and absolute lymphocyte count [117]. Additionally, the BM742401 methylation status was significantly associated with a higher white blood cell count and advanced Rai stage [118]. Table 3 summarizes the details of the important lncRNAs associated with CLL.

## 4. Alterations in lncRNAs in Non-Hodgkin Lymphoma

The term of “Non-Hodgkin lymphoma” (NHL) refers to a group of diverse lymphoid cell malignancies that can arise from mature or immature B lymphocytes, T lymphocytes, or natural killer (NK) cells [120]. According to the World Health Organization (WHO), the classification of NHL is based on immunophenotypic, genetic, and clinical characteristics. In the past decade, studies have shown that the dysregulation of lncRNAs plays a crucial role in chemotherapy response, survival, and apoptosis in lymphoma cells [121]. Here, we have we provide an overview of the discovered lncRNAs, as well as the mechanisms of action of these molecules in different types of NHLs.

### 4.1. Alterations in lncRNAs in B-Cell Lymphomas

The most common types of B-cell lymphomas include chronic lymphocytic leukemia (CLL), mantle-cell lymphoma (MCL), Burkitt’s lymphoma (BL), follicular lymphoma (FL), and diffuse large B-cell lymphoma (DLBCL). Several studies have identified differently expressed lncRNAs in B-cell NHLs by using datasets that confirm the findings in tissue samples or cell lines (Figure 3). For example, HAGLROS, MANCR, ROR1-AS1, and MALAT1 were found to be the most important upregulated lncRNAs in MCL patients [122]. While HAGLROS lead to the progress of MCL through regulation of PI3K/AKT/mTOR signaling [123], ROR1-AS1 physically interacts with core proteins of the PRC2 complex (especially EZH2) and suppresses the expression of SOX11 [124]. Wen et al. reported that downregulation of MANCR induces an antiproliferative effect in MCL, possibly by interacting with RUNX2 [125]. Wang et al. demonstrated that knockdown of MALAT1 in MCL cell lines leads to cell cycle arrest [126]. The differentially overexpressed lncRNAs in BL were found to be MCM3AP-AS1, MINCR, and NORAD. Guo et al. reported that MCM3AP-AS1 causes the expression of EIF4E and its downstream anti-apoptotic proteins by sponging miR-15a, which lead to the cell proliferation and chemoresistance [127]. Doose et al. demonstrated that MINCR act as a modulator of the MYC transcriptional program, and knockdown of this lncRNA is associated with an impairment in cell cycle progression in BL [128]. Similarly, Li et al. showed that NORAD is upregulated in the blood of patients with BL and has a diagnostic value [129]. On the other hand, FAS-AS1 led to an alternative splicing mechanism of Fas, inducing Fas-mediated apoptosis in lymphoma, which is downregulated in BL and DLBCL [130]. The most-studied lncRNA in FL, for which its biological functions are not yet clear, is RP11-625 L16.3 [131].

As the most common type of NHL, several differentially expressed lncRNAs have been identified in DLBCL. While SNHG14, OR3A4, NEAT1, FIRRE, MALAT1, HOTAIR, LUNAR1, SMAD5-AS1, and HULC are the most important upregulated lncRNAs, PANDA, FAS-AS1, lincRNA-21, and TUG1 are the most studied downregulated lncRNAs in DLBCL [122]. It has been reported that the lncRNA SNHG14 causes the evasion of malignant cells from the immune system by upregulating the expression of PD-L1 through sponging miR-5590-3p and Zinc finger Ebox binding homeobox 1 (ZEB1) [132]. Furthermore, some other lncRNAs, such as SMAD5-AS1, FIRRE and OR3A4, have been reported to increase the survival and proliferation of malignant cells, mainly through hyperactivation of the Wnt/b-catenin signaling pathway [95,133,134]. As another upregulated lncRNA in DLBCL patients, HOTAIR can induce the exposure of the promoter of target genes involved in MAPK/Erk signaling, promoting cell proliferation and predicting a poor prognosis in patients [135]. Similarly, It has been reported that the lncRNA MALAT1 can lead to H3K27me3 modification by interacting with the enhancer of zeste homolog 2 (EZH2), inducing its target genes in DLBCL and other malignancies [126]. Qian et al. reported that NEAT1 promotes lymphomagenesis and B-cell proliferation through a MYC-regulated mechanism in DLBCL [95]. Additionally, LUNAR1 can lead to the proliferation of malignant cells in DLBCL. Peng et al. reported that inhibition of this lncRNA can significantly induce cell cycle arrest and upregulation of p21 expression [136]. As another upregulated lncRNA which can be a possible therapeutic target for treatment of DLBCL patients, the lncRNA HULC can significantly increase the level of Bcl-2, and knockdown of HULC induced apoptosis of DLBCL cells [137]. Furthermore, decreased levels of the lncRNA PANDA were found in DLBCL cells. Zhou et al. reported that the lncRNA PANDA can negatively regulate the MAPK/Erk signaling pathway in DLBCL, and downregulation of this lncRNA leads to decreased tumor cell apoptosis and increased cell proliferation [138]. Additionally, downregulation of LincRNA-p21, which act as an tumor suppressor and regulates the proliferation of p53 signaling, is markedly decreased in DLBCL cells compared to normal cells [139].

### 4.2. Alterations in lncRNAs in T/NK-Cell Lymphomas

T and NK-cell lymphomas are a complex and heterogeneous group of neoplasms. Currently, there have been several reports of deregulation of lncRNAs in T and NK-cell lymphomas, but the precise mechanism of action of many of them is still unknown. LncRNA ZFAS1 overexpression in Natural killer/T-cell lymphoma (NKTCL) cases may encourage NK-cell proliferation through multiple mechanisms. First, ZFAS1 may be involved in NKTCL by deregulating P53-mediated pathways. There is a hypothesis that ZFAS1 may regulate P53 indirectly by regulating MDM2. Additionally, ZFAS1 may stimulate growth by deregulating different pathways that have been found to be overactive, such as the NF-B, WNT, or NOTCH1 pathways [140]. Kim et al. reported that lncRNA MALAT1 expression increased in T and NK-cell lymphomas, and high expression was found in patients with a worse prognosis. MALAT1 expression was upregulated, which increased PRC2 complex activity by interacting with the SUZ12 and EZH2 proteins. Additionally, the reported high expression of MALAT1 may cause BMI1 activation, which is linked to patients with a worse prognosis [141]. Mularoni et al. reported that the lncRNA MTAAT is related to the progression of aggressive ALK-negative anaplastic large cell lymphoma (ALK-ALCL). MTAAT has an intrinsic ability to regulate transcription, and loss of MTAAT leads to an increase in aberrant mitochondrial turnover through positive mitophagy stimulation, leading to reduced cell proliferation [142]. Recently, Fragliasso et al. reported that a chromatin-associated lncRNA, BlackMamba, can play a crucial role in the growth and morphology of ALK-ALCL cells, mainly by regulating the interaction between the DNA helicase HELLS and promoter regions of cell-architecture-related genes [143]. LncRNA MIR503HG improves the capacity of ALK-ALCL for growth by protecting and activating miR-503/Smuf2/TGFBR pathway components [144].

## 5. Future Prospectives and Clinical Applications of lncRNAs

In general, the applications of lncRNAs in the clinic can be divided into two different phases: as biomarkers and as therapeutic tools in targeted cancer therapies. As biomarkers, three methods, including RNA sequencing, custom microarrays, and quantitative RT-PCR, can be used to investigate the prognostic and predictive value of lncRNAs [145]. Some lncRNAs present in body fluids (such as blood, saliva, and urine) can be derived from apoptotic or necrotic cells or secreted as exosome content from living cells, which can be used as an easy and non-invasive detection method [146]. LncRNAs can be used as therapeutic tools to selectively kill tumor cells through various mechanisms. Oncogenic lncRNAs can be targeted by silencing their expression and inhibiting their activity by blocking their interaction. Delivery of synthetic oligonucleotide-based molecular products is one of the most researched methods to suppress upregulated oncogenic lncRNAs. This method offers simplicity in dosage management, minimal immunogenicity, and a lack of risk of genome integration. Small interfering RNAs (siRNAs) are complementary and antisense double-stranded RNA oligonucleotides that target lncRNAs. They do this by recruiting the RNA-induced silencing complex to cause the destruction of their target. Another approach involves targeting lncRNAs linked to chemoresistance with interference techniques, which may improve drug response and treatment outcomes [145]. However, a significant challenge remains in making the delivery more efficient and achieving a long-lasting effect in patients [147]. Common gene therapy approaches can be used to upregulate tumor suppressor lncRNAs. Additionally, genome editing strategies like CRISPR/Cas9 are being developed to knock out or knock in desired lncRNAs [148].

## 6. Conclusions

LncRNAs were once considered cellular waste until modern genomic platforms discovered that they are gene expression regulators. Now, we have a clearer understanding of how lncRNAs affect various complicated physiological processes. Recent advances in epigenomic and deep RNA-sequencing have revealed that these molecules are active participants in immune system responses. In this regard, given their find-tuned functions in lymphopoiesis, it is reasonable to assume that dysregulation of related lncRNAs (directly or indirectly regulating gene expression involved in molecular mechanisms such as cell proliferation and apoptosis) could trigger the onset of lymphoid leukemias, whether in the context of acute or chronic leukemia. Regarding the growing list of relevant lncRNAs in the pathogenesis and prognosis of lymphoid malignancies, the application of these biomarkers as early diagnostic indicators, prognostic markers, or novel therapeutic targets should be considered in future studies. Indeed, we believe that continued investigations into lncRNAs will yield new discoveries and reveal novel insights into better therapeutics for treating lymphoid leukemias.

## Figures and Tables

**Figure 1 ncrna-09-00044-f001:**
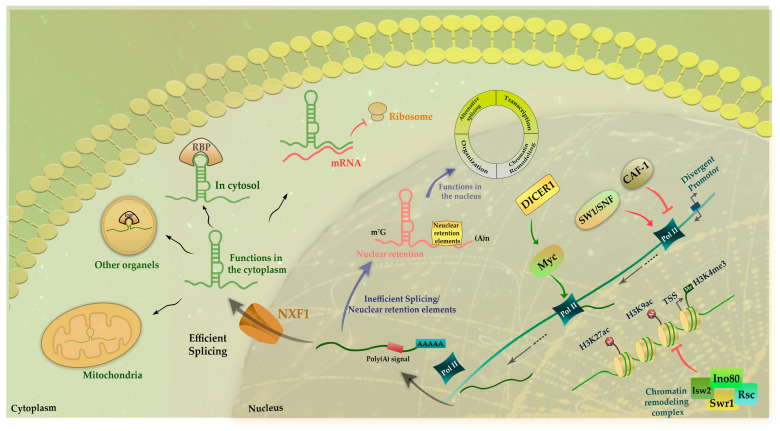
An overview of lncRNA biogenesis. At the chromatin level, lncRNA promoters have higher levels of H3K4me3, H3K27ac, and H3K9ac, which are more strongly repressed by chromatin remodeling complexes such as Swr1, Isw2, Rsc, and Ino80. The transcription of lncRNA starts when the SWI/SNF chromatin remodeling complex facilitate access of RNA polymerase II (Pol II) to the core promoter, while the elongation stage is controlled by MYC and DICER1, and its synthesis is complete by 3′-polyadenylation. Then, synthesized lncRNAs should be spliced, capped, and/or polyadenylated by recruiting proteins. This procedure is mediated by regions within the immature lncRNAs such as polyadenylation signal regions. Many Pol II-transcribed lncRNAs are not efficiently processed and are retained in the nucleus. Binding to the nuclear retention element is another mechanism leading to the retention of lncRNAs in the nucleus. Nuclear organization, regulation of gene transcription, alternative splicing, and chromatin remodulation are the main functions of lncRNAs in the nucleus. Other lncRNAs are spliced and exported to the cytoplasm by nuclear RNA export factor 1 (NXF1). In the cytoplasm, lncRNAs typically interact with a variety of RBPs. Some of them are sorted into mitochondria, some of them are associated with ribosomes, and others reside in organelles, such as exosomes. As one of the main mechanisms of action of lncRNA-mediated mRNA stability regulation, lncRNAs can directly bind to their target mRNAs and avoid interactions with ribosomes.

**Figure 2 ncrna-09-00044-f002:**
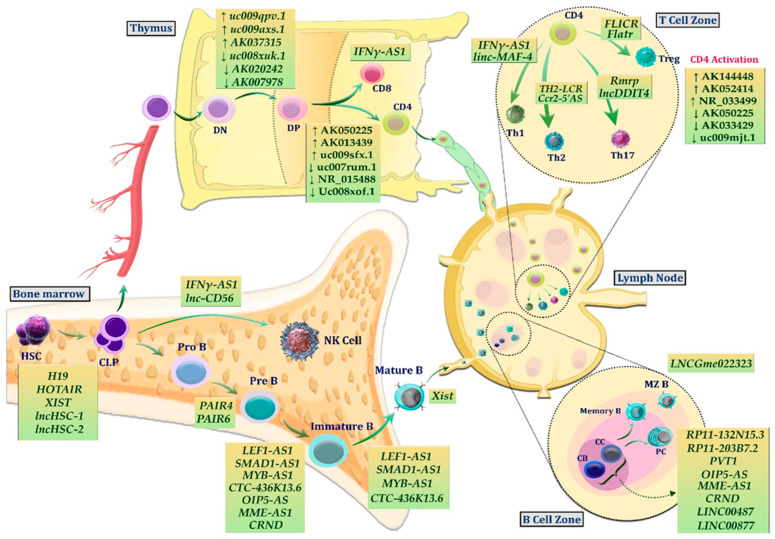
LncRNAs can regulate the differentiation of B and T lineages from hematopoietic stem cells into mature cells, as well as the subsequent activation of these lymphocytes. **HSC**: Hematopoietic stem cell; **CLP**: common lymphoid progenitor; **DN**: double negative; **DP**: double positive; **MZB**: marginal zone B cell; **CB**: centroblast; **CC**: centrocyte.

**Figure 3 ncrna-09-00044-f003:**
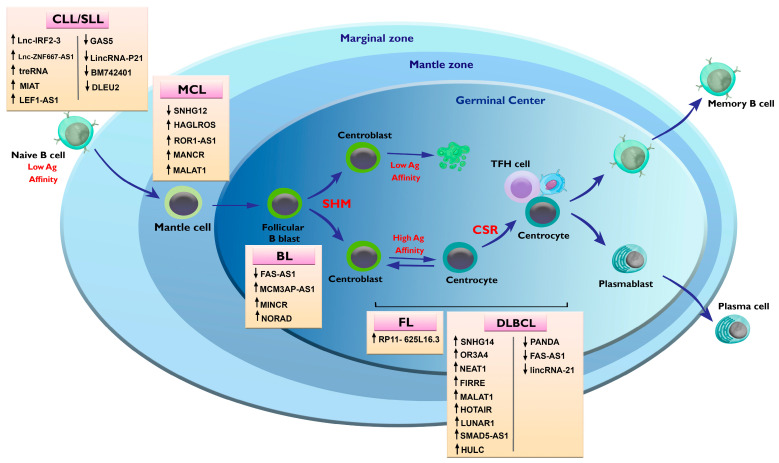
Alterations in lnsRNAs in different NHLs. **CLL/SLL**: Chronic lymphocytic leukemia/small lymphocytic lymphoma; **MCL**: mantle cell lymphoma; **BL**: burkitt lymphoma; **FL**: follicular lymphoma; **DLBCL**: diffuse large B-cell lymphoma; **SHM**: somatic hyper mutation; **CSR**: class switch recombination.

**Table 1 ncrna-09-00044-t001:** An overview of the most important lncRNAs associated with B-ALL.

	Alteration	Subtype	Target	Function	Ref.
**Oncogenic lncRNAs**					
**TCL6**	Upregulated	ETV6-RUNX1	Unknown	Low expression associated with poor disease-free survival	[71]
**LINC0098**	Upregulated	Pre B-ALL t(12:21) ETV6-RUNX1 and High hyperdiploid	miR-330-5p	miRNA sponge	[85]
**BALR-2**	Upregulated	B-ALL t (12;21), t (1;19) and MLL-rearrangement	JUN and BIM	Increases cell growth and prednisolone resistance, associated with poor survival	[74]
**RP11-137H2.4**	Upregulated	B-ALL	NRAS/BRAF/NF-κB MAPK pathways	Inhibits proliferation and migration	[80]
**ENST00000443469**	Upregulated	B-ALL MLL-rearrangement	LAMP5	Promotes cell proliferation and inhibits apoptosis	[82]
**LINC00152**	Upregulated	B-ALL	Unknown	Associated with risk of early relapse	[81]
**BALR-6**	Upregulated	B-ALL MLL-rearrangement	SP1-mediated transcription of CREB1	Promotes cell proliferation and inhibits apoptosis	[86]
**CRNDE**	Upregulated	B-ALL	miR-345-5p	Promotes cell proliferation and inhibits apoptosis	[87]
**ZEB1-AS1**	Upregulated	B-ALL	IL11/STAT3 pathway	Increases cell proliferation	[88]
**Tumor suppressive lncRNAs**					
**IUR**	Downregulated	B-ALL Ph+	STAT5 and CD71	Increases cell growth and survival	[75]
**LINC00221**	Downregulated	B-ALL	miR 152-3P and ATP2A2	Promotes apoptosis and inhibits tumor proliferation	[76]
**CASC15**	Upregulated	B-ALL ETV6-RUNX1	SOX4	Promotes apoptosis and inhibits tumor proliferation	[74,78]
**lnc-NKX2-3-1**	Upregulated	B-ALL ETV6-RUNX1	Unknown	Unknown	[72]
**lnc-TIMM21-5**	Upregulated	B-ALL ETV6-RUNX1	Unknown	Unknown	[72]
**lnc-ASTN1-1**	Upregulated	B-ALL ETV6-RUNX1	Unknown	Unknown	[72]
**lnc-RTN4R-1**	Upregulated	B-ALL ETV6-RUNX1	Unknown	Unknown	[72]

**TCL6:** T-cell leukmia/lymphoma 6; **BALR-2:** B-ALL associated long RNA-2; **LINC0098:** long intergenic non-protein coding RNA 98; **BALR-6:** B-ALL associated long RNA-6; **CRNDE:** colorectal neoplasia differentially expressed; **ZEB1-AS1:** zinc finger E-box binding homeobox 1 antisense RNA 1; **IUR:** imatinib-upregulated lncRNA; **LINC00221:** long intergenic non-protein coding RNA 221; **CASC15:** cancer susceptibility candidate 15; **lnc-TIMM21-5:** long non-coding RNA translocase of inner mitochondrial membrane 21-5; **lnc-ASTN1-1:** long non-coding RNA astrotactin-1; **lnc-RTN4R-1:** long non-coding RNA reticulon 4 receptor-1.

**Table 2 ncrna-09-00044-t002:** An overview of the most important lncRNAs associated with T-ALL.

	Alteration	Target	Function	Ref.
**SNHG1**	Upregulated	miR 124-3P	Promotes cell proliferation and migration	[107]
**PVT1**	Upregulated	miR-486-5p, NOP2 and Myc	Increases cell viability, deregulates cell cycle, and inhibits apoptosis	[97]
**ARIEL**	Upregulated	ARID5B	Increases cell growth and survival	[93]
**LUNAR1**	Upregulated	IGF1R	Increases cell growth	[99,108]
**NALT**	Upregulated	Notch signaling	Induces cell proliferation	[100]
**NEAT1**	Upregulated	miR-146b-5p	Promotes cell proliferation and growth	[101]
**AWPPH**	Upregulated	ROCK2	Promotes cell proliferation and inhibits apoptosis	[104]
**T-ALL-R-LncR1**	Upregulated	Par-4/THAP1 protein complex	Inhibits apoptosis	[105]
**H19**	Upregulated	SOX2, OCT4 and NANOG	Maintains stemness, promotes cell proliferation, and inhibits apoptosis	[102,109]
**LINC00853**	Downregulated	CCR9 expression	Inhibits cell proliferation and migration	[110]
**EBLN3P**	Upregulated	miR-655-3p	Increases proliferation, invasion, and migration	[106]
**PPM1A-AS**	Upregulated	Notch, PI3K-AKT, and JAK-STAT signaling	Regulates cell proliferation and apoptosis	[111]
**LINC00478**	Upregulated	mir-125b	Increases cell growth and invasiveness	[112]

**SNHG1:** small nucleolar RNA host gene 1; **PVT1:** plasmacytoma variant translocation 1; **ARIEL:** ARID5B inducing enhancer associated long noncoding RNA; **LUNAR1:** leukemia-induced noncoding activator RNA; **NALT:** NOTCH1 associated lncRNA in T-ALL; **NEAT1:** nuclear enriched abundant transcript 1; **AWPPH:** long non-coding RNA associated with poor prognosis of hepatocellular carcinoma; **T-ALL-R-LncR1:** T-ALL-related long non-coding RNA; **LINC00853:** long intergenic non-protein coding RNA 853; **EBLN3P:** endogenous bornavirus-like nucleoprotein; **PPM1A-AS:** protein phosphatase 1A antisense RNA; **LINC00478:** long intergenic non-protein coding RNA 478.

**Table 3 ncrna-09-00044-t003:** An overview of the most important lncRNAs associated with CLL.

	Alteration	Target	Function	Ref.
**Lnc-IRF2-3**	Upregulated	Unknown	Promotes cell proliferation and inhibits apoptosis	[113]
**Lnc-ZNF667-AS1**	Upregulated	Unknown	Poor prognosis and survival	[113]
**treRNA**	Upregulated	ZAP70	Poor prognosis and survival	[114]
**MIAT**	Upregulated	OCT4	Inhibits apoptosis and increases aggressiveness	[115]
**GAS5**	Downregulated	miR-222	Promotes tumorigenesis and metastasis	[116]
**LincRNA-P21**	Downregulated	p53	Inhibits apoptosis	[117]
**BM742401**	Downregulated	Cas9 and 8	Promotes cell proliferation and inhibits apoptosis	[118]
**DLEU2**	Downregulated	miR-15a and miR-16-1	A tumor suppressor	[37]
**LEF1-AS1**	Upregulated	LEF1	Inhibits apoptosis	[119]

**Lnc-IRF2-3:** long non-coding RNA interferon regulatory factor 2-3; **Lnc-ZNF667-AS1:** LncRNA zinc finger 677 antisense RNA 1; **treRNA:** translational regulatory lncRNA; **MIAT:** Myocardial infarction associated transcript; **GAS5:** Growth arrest-specific 5; **LincRNA-P21:** long intergenic noncoding RNA p21; **DLEU2:** deleted in lymphocytic leukemia 2; **LEF1-AS1:** lymphoid enhancer-binding factor 1-antisense RNA 1.

## Data Availability

Not applicable.

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
