# Peer review of "Long Non-Coding RNA Signatures in Lymphopoiesis and Lymphoid Malignancies"

_ncrna, 2023, doi:10.3390/ncrna9040044_

Round 1
Reviewer 1 Report
This work provides a comprehensive overview of the current understanding of lncRNAs and their regulatory roles in lymphopoiesis as well as lymphoid leukemia. The authors effectively highlight the importance of studying the genetic transcriptome of lymphoid cells and shed light on the previously unrecognized regulatory potential of lncRNAs in these processes.
The paper provides a valuable synthesis of existing knowledge by collating relevant studies and research findings. It effectively communicates the current state of research in the field and highlights the potential implications of lncRNAs in the diagnosis, prognosis, and treatment of lymphoid malignancies. However, it would have been beneficial if the review had included a discussion of different mechanisms of how lncRNAs function, providing some case studies to help the audience better understand the regulatory roles of lncRNAs in lymphopoiesis and lymphoid leukemia.
Line 45-46: By definition, ncRNAs are functional single-stranded RNA molecules that are not translated into proteins, although there are several cases where ncRNAs are translated into peptides. Therefore, it is better to state: "Non-coding RNAs (ncRNAs) are functional single-stranded RNA molecules that are not translated into proteins, although there are exceptions where ncRNAs are translated into peptides."
Line 75: "Translated" is not the correct term here. Please replace it with "transformed."
Line 79-80: The statement "The transcription of lncRNA starts when tyrosine1 is phosphorylated in Pol II" is not accurate, as not all Pol II is phosphorylated in lncRNA transcription, although most of it is.
Figure 1: In the cytoplasm, lncRNAs regulate translation by associating with ribosomes or mRNA. The figure drawing shows a ribosome translating lncRNA, which is misleading, and the ribosome label is on the wrong side. Although there are reported cases where lncRNA binds to ribosomes with a 5' UTR-like region to regulate the turnover of ribosomes, this cannot represent a typical regulatory role of lncRNAs in translation.
Sections 2-3: It would be better to explain more specifically how lncRNAs regulate these genes (chromatin, transcription, splicing, etc.) rather than just listing them.
Easy to read
Reviewer 2 Report
In this review manuscript, Baghdadi et al., reviewed the lncRNAs in lymphopoiesis and lymphoid leukemias. Some comments are below.
1. It’s unclear why the authors focus on lymphoid leukemias only but not expand to lymphomas and NK leukemia? In particular, the authors reviewed the lncRNAs in normal B/T/NK-lymphopoiesis and the subtitle of Section 3 is ‘Alterations of lncRNAs in lymphoid malignancies’, which should include lymphomas.
2. A more detail perspective about future investigation and clinical practice would be very fascinating.
3. It also would improve this review quite a lot if the authors could review more about lncRNAs and immunotherapy.
4. Line 55 to 76 should be one paragraph. It’s not necessary to describe a lot of sncRNA. The introduction is too tedious.
5. What’s the red box in lncRNA in Figure 1? No information of this red box is described.
6. I like the summary Figure 2. In Figure 2, the IFNγ-AS1 lncRNA was both showed in CD4+ and CD8+ T cells, but no description about it in CD8+ T cells. Any evidence showing lncRNAs in regulation of CD8+ T cell differentiation?
7. ncRNAs should also include circular RNAs, right?
8. Line 280 – 281, lncRNAs should not be tiny molecules, particularly some are quite large. Line 281, ‘some lncRNA’ should be ‘some lncRNAs’. The authors should carefully check the grammars throughout.
There are some grammar issues. The authors should carefully check the grammars throughout.
Round 2
Reviewer 2 Report
The authors addressed my concerns except multiple myeloma should not be considered as a NHL or B-cell lymphoma. Line 509-528 should be deleted. Same, the summary figure should be revised.
Author Response
Thanks for your exact point of view. Multiple myeloma is a plasma cell neoplasms subgroup; we removed this section and edited Figure 3.